# Integrative Identification of Genetic Loci Jointly Influencing Diabetes-Related Traits and Sleep Traits of Insomnia, Sleep Duration, and Chronotypes

**DOI:** 10.3390/biomedicines10020368

**Published:** 2022-02-02

**Authors:** Yujia Ma, Zechen Zhou, Xiaoyi Li, Zeyu Yan, Kexin Ding, Han Xiao, Yiqun Wu, Tao Wu, Dafang Chen

**Affiliations:** Department of Epidemiology and Biostatistics, School of Public Health, Peking University, Beijing 100191, China; 2111110185@pku.edu.cn (Y.M.); peevesomega@163.com (Z.Z.); lvxi520@163.com (X.L.); 1510306132@pku.edu.cn (Z.Y.); 1610306225@pku.edu.cn (K.D.); gesangmeiduo@pku.edu.cn (H.X.); qywu118@163.com (Y.W.); twu@bjmu.edu.cn (T.W.)

**Keywords:** type 2 diabetes, genetic pleiotropy, genome-wide association study, sleep

## Abstract

Accumulating evidence suggests a relationship between type 2 diabetes mellitus and sleep problems. A comprehensive study is needed to decipher whether shared polygenic risk variants exist between diabetic traits and sleep traits. Methods: We integrated summary statistics from different genome-wide association studies and investigated overlap in single-nucleotide polymorphisms (SNPs) associated with diabetes-related traits (type 2 diabetes, fasting glucose, fasting insulin, and glycated hemoglobin) and sleep traits (insomnia symptoms, sleep duration, and chronotype) using a conditional/conjunctional false discovery rate approach. Pleiotropic genes were further evaluated for differential expression analysis, and we assessed their expression pattern effects on type 2 diabetes by Mendelian randomization (MR) analysis. Results: We observed extensive polygenic pleiotropy between diabetic traits and sleep traits. Fifty-eight independent genetic loci jointly influenced the risk of type 2 diabetes and the sleep traits of insomnia, sleep duration, and chronotype. The strongest shared locus between type 2 diabetes and sleep straits was *FTO* (lead SNP rs8047587). Type 2 diabetes (z score, 16.19; P = 6.29 × 10^−59^) and two sleep traits, sleep duration (z score, −6.66; P = 2.66 × 10^−11^) and chronotype (z score, 7.42; P = 1.19 × 10^−13^), were shared. Two of the pleiotropic genes, *ENSA* and *PMPCA*, were validated to be differentially expressed in type 2 diabetes, and *PMPCA* showed a slight protective effect on type 2 diabetes in MR analysis. Conclusions: Our study provided evidence for the polygenic overlap between diabetic traits and sleep traits, of which the expression of *PMPCA* may play a crucial role and provide support of the hazardous effect of being an “evening” person on diabetes risk.

## 1. Introduction

Type 2 diabetes mellitus is a complex disease induced by a combination of environmental and genetic factors. It is estimated that approximately 463 million adults aged 20–79 years suffer from diabetes globally, which is expected to surge to 700 million by 2045 [1,2]. Complications of diabetes seriously affect the physical health of patients and lead to a heavy health burden of disability and mortality, consuming massive loss of social resources [1]. Genome-wide association studies (GWAS) have identified more than 500 susceptibility loci that demonstrate a robust association with type 2 diabetes [3]. In contrast to the tremendous stride in GWAS research, the conundrum of “missing heritability” in type 2 diabetes has progressed slowly and arduously. The identified genetic variants explain only 19% of the familial clustering of type 2 diabetes [4,5].

An extensive overview of pleiotropy and genetic architecture showed that 90% of trait-associated loci overlap with loci from multiple traits [6]. Combining GWAS from multiple phenotypes provides insights into genetic pleiotropy and could elucidate shared pathobiology [7]. The conjunctional false discovery rate (conjFDR), an extension of the conditional false discovery rate (condFDR), is such an approach that boosts GWAS discovery by leveraging auxiliary genetic information to readjust the GWAS test statistics in a primary phenotype and was applied for cross-trait analysis by leveraging overlapping SNP associations between separate GWAS to rerank the test statistics in a primary phenotype conditional on the associations in a secondary phenotype [8,9]. This method is a model-free strategy for the analysis of GWAS summary statistics inspired by the empirical Bayes statistical framework, which is designed for situations with dense elements, such as the large number of small genetic effects seen in polygenic traits and disorders [8,9].

Accumulating evidence suggests that sleep traits may have indispensable effects on the development of type 2 diabetes, such as insomnia and chronotype. Insomnia disorder is the second-most prevalent mental disorder with prevalence estimates ranging from 10% (adults) to 22% (elderly) and is characterized by lasting problems falling asleep or waking up in the night or early morning, with subjective repercussions for daytime functioning [10]. The adverse effect of insomnia on type 2 diabetes risk was verified by multiple observational studies and Mendelian randomization studies [11,12,13]. A 12-day inpatient General Clinical Research Center study found that sleep restriction significantly reduces insulin sensitivity [14], and simple sleep interventions such as sleep extension are associated with improvements in fasting insulin sensitivity [15]. In addition to the above epidemiological evidence, genome-wide association studies (GWAS) have provided new insights into the complex genetic mechanisms between type 2 diabetes and sleep traits. Polygenic risk scores for sleep duration obtained from GWAS summary statistics are associated with an increased likelihood of various metabolic traits [16]. There is also a correlation between genetic risk factors for insomnia and the risk of type 2 diabetes (r_g_ = 0.20) [17]. Chronotype of an individual refers to the specific entrainment and/or activity-rest preference of that individual in a given 24-h day [18]. It can be denoted as circadian topology or diurnal preference and may manifest as measures of the timing of actual sleep-wake behaviors or preference for sleep-wake timing under idealized conditions [19]. Early risers who are preferentially active in the mornings are said to have a morning chronotype and are often dubbed as larks, and late risers with more nocturnal activities have late chronotypes and are popularly dubbed owls. The literature suggests that circadian rhythms are important to weight regulation and metabolism. Suggested mechanisms include dietary behavior, appetite-stimulating hormones, and glucose metabolism [20]. Therefore, shared genetic influences of sleep traits can be highly valuable for type 2 diabetes to provide biological insights and uncover shared biological underpinnings. A comprehensive study is needed to decipher whether shared polygenic risk variants exist between diabetic traits and sleep traits, which is essential to unveil the genetic mechanisms of type 2 diabetes and impel early prevention and therapy.

In this study, we investigated the polygenic overlap between type 2 diabetes and sleep traits using the conjFDR approach and focused on pleiotropic genes. In order to better understand type 2 diabetes pathophysiology, we also included other diabetes-related traits, including fasting glucose (FG), glycated hemoglobin (HbA1c), and fasting insulin (FI). We further assessed whether the pleiotropic genes were enriched in particular pathways and their expression pattern effects on type 2 diabetes.

## 2. Materials and Methods

### 2.1. Study Participants

GWAS results in the form of summary statistics on type 2 diabetes were acquired from Mahajan et al.’s work [21]. In this study, 403 independent association signals were detected by conditional analyses at each of the genome-wide significant risk loci for type 2 diabetes (except at the major histocompatibility complex (MHC) region). Summary-level data are available at the DIAGRAM consortium (http://diagram-consortium.org/, accessed on 13 November 2020). European-specific meta-analysis summary-level results for fasting glucose (FG), glycated haemoglobin (HbA1c), and fasting insulin data (FI) were acquired from a trans-ancestral meta-analysis, which aggregated genome-wide association studies comprising up to 281,416 individuals without diabetes (70% European ancestry) [22]. European-specific meta-analysis summary-level results were downloaded through the MAGIC website (https://www.magicinvestigators.org/, accessed on 13 November 2020) and used for subsequent analysis.

Summary statistics results of sleep traits were obtained from Jansen et al.’s study [23]. The freely available meta-analytic sleep traits (insomnia symptoms, sleep duration, and chronotype) represent results partly provided by the UK Biobank Study (www.ukbiobank.ac.uk, accessed on 13 November 2020) [24].

The UK Biobank collected a single self-reported measure at baseline of sleep traits. Insomnia symptoms were assessed by asking, “Do you have trouble falling asleep at night or do you wake up in the middle of the night?”, with responses “Never/rarely”, “Sometimes”, “Usually”, or “Prefer not to answer”. Those who responded “prefer not to answer” were missing. Insomnia cases (*n* = 109,402) were defined as participants who answered this question with “usually”, while participants answering “never/rarely” or “sometimes” were defined as controls (*n* = 277,131). “Usually have trouble falling asleep at night or waking up in the middle of the night” may be the most important part of the Diagnostic and Statistical Manual of Mental Disorders, Fifth Edition (DSM-5) and International Classification of Sleep Disorders (ICSD) diagnostic criteria for insomnia disorder, so this definition of insomnia symptoms from the self-reported measure was validated to be closer to the DSM-5 and ICSD diagnostic criteria than the commonly used Insomnia Severity Index (ISI) or Pittsburgh Sleep Quality Index (PSQI). Additionally, it previously showed excellent sensitivity (98%) and specificity (96%) of the UK Biobank insomnia phenotype to differentiate between cases that consistently met both the ISI and PSQI criteria versus controls that consistently were below both the ISI and PSQI cut-off scores [10]. Thus, we used this phenotype as a proxy for insomnia. Sleep duration, obtained from 384,317 individuals, was a quantitative variable assessed by asking, “About how many hours sleep do you get in every 24 h? (please include naps)”. Chronotype (“Morning/evening person (chronotype)”; data-field 1180, *n* = 345,552) was assessed by the question “Do you consider yourself to be?” with one of six possible answers: “Definitely a ‘morning’ person”, “More a ‘morning’ than ‘evening’ person”, “More an ‘evening’ than a ‘morning’ person”, “Definitely an ‘evening’ person”, “Do not know”, or “Prefer not to answer”, which were coded as 2, 1, −1, −2, 0, and missing, respectively. Summary-level data are available at https://ctg.cncr.nl/software/summary_statistics (accessed on 13 November 2020).

### 2.2. Statistical Analysis

#### 2.2.1. Conditional Quantile–Quantile (Q–Q) Plots

We constructed conditional Q–Q plots to assess pleiotropic enrichment between diabetes-relevant traits and sleep traits. Conditional Q–Q plots compare the association with the primary phenotype (e.g., type 2 diabetes) across all single-nucleotide polymorphisms (SNPs) and within SNPs stratified by their association with the secondary phenotype (e.g., insomnia). Successive leftward deflections from the null distribution of conditional Q–Q plots denoted the existence of pleiotropic enrichment. Spurious enrichment was controlled after random pruning by selecting one random SNP per linkage disequilibrium (LD) block (defined by LD r^2^ > 0.1) averaged over 100 iterations.

#### 2.2.2. Identification for Pleiotropic Loci

We identified specific loci jointly involved with diabetes-relevant traits and sleep traits according to a condFDR statistical framework (https://github.com/precimed/pleiofdr) (accessed on 25 August 2021) [25]. CondFDR is an extension of the standard FDR. It incorporates information from GWAS summary statistics of a secondary phenotype to adjust its significance level. We denoted the condFDR for phenotype 1 given phenotype 2 as FDRtrait1|trait2, which is defined as the posterior probability that a given SNP is null for the first phenotype given that the *p*-values for both phenotypes are as small as or smaller than the observed ones. Based on CondFDR, we computed the conjunctional false discovery rate (conjFDR), denoted as FDRtrait1&trait2, the conservative estimate of which was given by the maximum between FDRtrait1|trait2 and FDRtrait2|trait1. It is defined as the posterior probability that an SNP is null for either phenotype or both simultaneously, given that its *p*-values for associations with both phenotypes are as small as or smaller than the observed ones. SNPs with a conjFDR value less than 0.01 were considered shared loci. Based on the 1000 Genome Project LD structure, the significant SNPs identified were clustered into LD blocks at the LD r^2^ > 0.1 level.

#### 2.2.3. Functional Annotation

The significant SNPs identified were annotated by SNPNexus (https://www.snp-nexus.org/v4/) (accessed on 26 August 2021) [26]. SNPs were annotated for functional consequences on deleteriousness score (CADD score) and potential regulatory functions (RegulumeDB score) [27,28]. A CADD score above 12.37 is the threshold to be potentially pathogenic [27]. The RegulumeDB score is based on information from eQTLs and chromatin marks, ranging from 1a to 7, with lower scores indicating an increased likelihood of having a regulatory function. In order to clarify the biological mechanism behind the pleiotropic genes, we conducted pathway enrichment analysis in the Kyoto Encyclopedia of Genes and Genomes (KEGG) dataset [29].

#### 2.2.4. Expression Analysis of Pleiotropic Genes

In order to evaluate whether the identified pleiotropic genes are differentially expressed, we used the publicly available expression dataset GSE184050 from the Gene Expression Omnibus (https://www.ncbi.nlm.nih.gov/geo/) (accessed on 13 September 2021) database. GSE184050 compared changes in gene expression using two longitudinally collected blood samples from subjects who transitioned to type 2 diabetes between the time points against those who did not, with a novel analytical network approach. A total of 116 individual samples (50 from type 2 diabetes cases and 66 from healthy controls) were submitted to the analysis. RNA was extracted, amplified, reverse transcribed, labeled, and sequenced with an Illumina HiSeq 2000 (GPL11154). We scaled the original data and deleted outliners defined as more than 3 standard deviations and used a Benjamini–Hochberg multiple-testing correction with a *p*-value < 0.05.

#### 2.2.5. Mendelian Randomization Study

In order to investigate causal associations between the expression pattern of pleiotropic genes and type 2 diabetes, we used eQTLGen 2019 results comprising all cis and some trans regions of gene expression in whole blood to perform a two-sample Mendelian randomization study. The eQTLGen consortium was set up to identify the downstream consequences of trait-related genetic variants. The consortium incorporates 37 datasets, with a total of 31,684 individuals [30]. We outlined acceptable instrumental variables via three main assumptions: they were associated with the relevant risk factor (relevance assumption), they and the outcome had no common cause (independence assumption), and the outcome was not affected by them except via the risk factor (exclusion restriction assumption) [31]. Genetic instrumental variables for eQTL summary statistics of pleiotropic genes were acquired from OpenGWAS, developed by the MRC IEU OpenGWAS project, the contributor of TwoSampleMR (https://github.com/mrcieu/TwoSampleMR) (accessed on 16 October 2021) package and MR-base [32]. The data setup of the open-access OpenGWAS database is scalable, open-source, high-performance, and cloud-based, importing and publishing complete GWAS metadata and summary datasets for scientific society. The import pipeline matches these datasets to the reference sequence of the human genome, and dbSNP produces summary reports and systematizes the results and metadata formats.

We used the widely accepted inverse-variance weighted (IVW) method for the main analysis to estimate the causal effect between pleiotropic genes and type 2 diabetes. The IVW estimate is calculated by regressing the coefficient from an outcome regression on the genetic variant on that from an exposure regression on the variant and weighting each estimate by the inverse variance of the association between the instrument and the outcome [33].

## 3. Results

### 3.1. Assessment of Pleiotropic Enrichment

We observed successive increments of SNP enrichment for diabetes-relevant phenotypes as a function of the significance of the associations with sleep traits (Figure 1). For a given nominal *p*-value for each diabetes-relevant trait, an earlier departure from the null line indicates a greater proportion of true associations. Gradual leftward shifts for decreasing nominal sleep traits *p*-values indicate that the proportion of nonnull SNPs varies considerably across different levels of association with sleep traits, which could be interpreted as the polygenic overlap between these phenotypes. Type 2 diabetes showed obvious pleiotropic enrichment with sleep traits. All diabetes-relevant phenotypes showed significant pleiotropy with chronotype.

### 3.2. Pleiotropic Gene Loci in Diabetes-Relevant Phenotype and Sleep Traits Identified with ConjFDR

Based on a conjFDR less than 0.05, we identified 58 independent genetic loci shared between type 2 diabetes and sleep traits (Figure 2). For FG, FI, and HbA1c, 22, 8, and 11 independent genetic loci were shared with sleep traits (Table 1; Appendix A). The strongest shared locus between type 2 diabetes and sleep straits was *FTO* (lead SNP rs8047587). It was shared between type 2 diabetes (z score, 16.19; P = 6.29 × 10^−59^) and two sleep traits, sleep duration (z score, −6.66; P = 2.66 × 10^−11^) and chronotype (z score, 7.42; P = 1.19 × 10^−13^), demonstrating the importance of the locus for disease pathogenesis. Two loci, EHMT2 (lead SNP rs1265945) and lincRNA RP1-230L10.1 (lead SNP rs66930764), shared by type 2 diabetes and chronotype, were duplicated in FI. The pleiotropic locus *MTNR1B* (lead SNP rs4237555) was identified among type 2 diabetes (z score, −7.86; P = 3.84 × 10^−15^), FG (z score, −19.19; P = 4.20 × 10^−82^), HbA1c (z score, −7.50; P = 6.26 × 10^−14^), and chronotype (z score, −4.85; P = 1.22 × 10^−6^).

### 3.3. Functional Annotation of Pleiotropic Gene

Five SNPs (rs10881959, rs11039358, rs2236950, rs12485697, rs1296328) had CADD scores greater than 12.37, suggesting that they might be deleterious mutations (Appendix A). One SNP (rs174555), shared among FG, HbA1c, and sleep duration, had Regulome DB scores of 1f, indicating that it was likely affecting binding sites (Appendix A). At the false discovery rate 0.05 level, KEGG pathway enrichment analysis found that *HSD17B12*, *FADS2*, and *FADS1* were significantly enriched in the biosynthesis of unsaturated fatty acids (hsa01040), of which *FADS2* and *FADS1* were the overlapping genes with SNP rs174555.

### 3.4. Differential Expression of Pleiotropic Genes

Among the pleiotropic genes screened, we found 12 genes differentially expressed in blood samples of type 2 diabetes cases (*p* < 0.05, Appendix A). *ENSA* and *PMPCA* remained significant after the stringent statistical analysis using the Benjamini–Hochberg corrected two-tailed *t*-test (Appendix A). *ENSA* (lead SNP rs2055975), overexpressed in type 2 diabetes cases, was shared by HbA1c (z score, −5.15; P = 2.57 × 10^−7^) and chronotype (z score, 5.73; P = 1.00 × 10^−8^). *PMPCA* (lead SNP rs10747046), which was downregulated in type 2 diabetes cases, was shared between FG (z score, −4.32; P = 1.53 × 10^−5^) and chronotype (z score, 4.47; P = 7.79 × 10^−6^).

### 3.5. Mendelian Randomization Study

In a two-sample MR study, IVW yielded proof of causal relationships between the expression level of pleiotropic genes and the risk of type 2 diabetes (Table 2; Appendix A). Overexpression of *CPEB3*(OR = 1.43, 95% CI: 1.30–1.56, *p* < 0.0001), *INPP5E* (OR = 1.10, 95% CI: 1.07–1.13, *p* < 0.0001), and *SEC16A* (OR = 1.08, 95% CI: 1.05–1.12, *p* < 0.0001) were associated with higher risk for developing type 2 diabetes, while *MYBPC3*(OR = 0.95, 95% CI: 0.92–0.99, *p* = 0.0173), *MYRF*(OR = 0.94, 95% CI: 0.90–0.98, *p* = 0.0049), and *PMPCA* (OR = 0.74, 95% CI: 0.62–0.87, *p* = 0.0003) showed slightly protective effect on type 2 diabetes (Table 2). The weighted medians for *MYBPC3* and *INPP5E* revealed similar estimates. Intriguingly, both *PMPCA* and *INPP5E* showed a significant association with chronotype, which is in opposite directions to type 2 diabetes, indicating that people who are prone to be more an “evening” than a “morning” person have a higher risk for developing type 2 diabetes. IVW yielded an association between chronotype and type 2 diabetes (OR = 1.37, 95% CI: 1.09–1.72, *p* = 0.0068), while other estimates showed that it was not robust. We also considered the causal effect of the differentially expressed genes *ENSA* and *PMPCA* on the shared phenotypes. Wald’s ratio method estimated that the overexpression of *PMPCA* had slight effects on lowering FG (OR = 0.90, 95% CI: 0.86–0.94, *p* < 0.0001).

## 4. Discussion

In the current study, we observed extensive polygenic pleiotropy between diabetic traits and sleep traits using conjFDR analysis. Fifty-eight independent genetic loci jointly influenced the risk of type 2 diabetes and the sleep traits of insomnia, sleep duration, and chronotype. Two of the pleiotropic genes, *ENSA* and *PMPCA*, were validated to be differentially expressed in type 2 diabetes, and *PMPCA* showed a slight protective effect on type 2 diabetes in MR analysis. Our study provides integrative evidence of a shared genetic mechanism between diabetes and sleep traits.

The strongest shared locus between type 2 diabetes and sleep traits was *FTO* (lead SNP rs8047587), a well-known gene associated with body mass index, obesity risk, and type 2 diabetes. However, the association between *FTO* and sleep traits has not been well discerned. Prats-Puig et al. showed that TT homozygotes for the *FTO* SNP exhibited nominal associations between decreasing sleep duration and increasing BMI, waist circumference, visceral fat, and HOMA-IR (all *p* < 0.05) in 297 asymptomatic children aged 5–9 years [34]. It is worth noting that *FTO* is predominantly expressed in the brain. Disruption in mice of Fto showed diet- or obesity-related changes in expression in the hypothalamus [35,36]. Abundant evidence supports multiple possible roles of the central nervous system in body weight regulation [37], and our study emphasized the role of sleep in the regulatory process.

Two notable pleiotropic genes were *ENSA* (lead SNP rs2055975) and *PMPCA* (lead SNP rs10747046), which were differentially expressed in type 2 diabetes cases. *ENSA* is expressed in brain and endocrine tissues and was proposed as a candidate gene for type 2 diabetes. It encodes alpha-endosulfine, which has the ability to block ATP-sensitive potassium (K(ATP)) channels and stimulate insulin release in beta cells such as sulfonylurea [38]. The considerably decreased alpha-endosulfine could result in a decrease in neurotransmitter release associated with cognition [38]. In our study, *PMPCA* showed a slight protective effect on type 2 diabetes and lowered FG. The literature on the direct role of *PMPCA* in diabetes is sparse, while a homozygous mutation in *PMPCA* has been reported to be crucial for autosomal recessive cerebellar ataxia [39,40]. *PMPCA* encodes the α-subunit of mitochondrial processing peptidase (MPP), a heterodimeric enzyme responsible for the cleavage of nuclear-encoded mitochondrial precursor proteins after import into mitochondria [41]. As previously mentioned, mitochondrial dysfunction leads to impairment of insulin sensitivity by reducing the activity of AMPK, an important cellular fuel sensor and regulator [42]. Agents addressing impaired mitochondrial function were thought to have the greatest potential for supporting a substantial improvement of glycemic and body weight control in the growing population with type 2 diabetes [43]. This may partly explain the pleiotropy of *PMPCA* in type 2 diabetes and sleep traits.

Our study showed that both *PMPCA* and *INPP5E* showed a significant association with chronotype, which is in opposite directions with type 2 diabetes, which suggested that people who are prone to be more of an “evening” than a “morning” person have a higher risk for developing type 2 diabetes. This is consistent with the latest systematic review and a cross-sectional study showing that evening chronotype was associated with a worse cardiometabolic risk profile and a higher risk of diabetes, cancer, and depression [44,45]. The latest research showed circadian rhythm disruption perturbed glucose homeostasis through disruption of pancreatic β cell function and loss of circadian transcriptional and epigenetic identity [46]. However, the opposite result was found in MR analysis, in which the IVW estimate yielded a morning chronotype and had an adverse effect on type 2 diabetes (OR = 1.37, 95% CI: 1.09–1.72, *p* = 0.0068). On the one hand, the odd results of the unrobust MR analysis suggest that MR studies should be validated more widely with multiple methods. On the other hand, Reis-Canaan’s study showed that most morning chronotype individuals were elderly thin males with lower consumption of omega-6 and omega-3, sodium, zinc, thiamine, pyridoxine, and niacin, whereas evening individuals were younger, had higher BMI, and had higher consumption of the studied micronutrients [47]. This indicates that the association between diabetes and chronotype is extremely entangled. The interpretation should be careful, and further well-designed studies should be conducted.

Our research had some limitations. First, overlapping participants between the investigated GWAS may inflate the cross-trait enrichment in the condFDR statistical framework. However, we had to choose a stringent threshold (conjFDR < 0.01) instead of the default parameter setting (0.05) to control for false positives. Another limitation is the use of self-reported sleep symptoms rather than clinical diagnostic criteria. Measurement errors and recall bias would result in misclassification of case status, especially for insomnia which we used insomnia complain as a proxy. Although a previous study showed that the UK Biobank insomnia phenotype is predictive of insomnia disorder, with little confounding by comorbidity [10], large-scale summary statistics for a precise definition of clinical diagnostic insomnia was desired in subsequent studies. Finally, our study requires more fundamental work to detect the underlying biological mechanisms between diabetes and sleep traits.

## 5. Conclusions

Our study provided evidence for the polygenic overlap between diabetic traits and sleep traits, of which the expression of *PMPCA* may play a crucial role and provide support of the hazardous effect of being an “evening” person on diabetes risk.

## Figures and Tables

**Figure 1 biomedicines-10-00368-f001:**
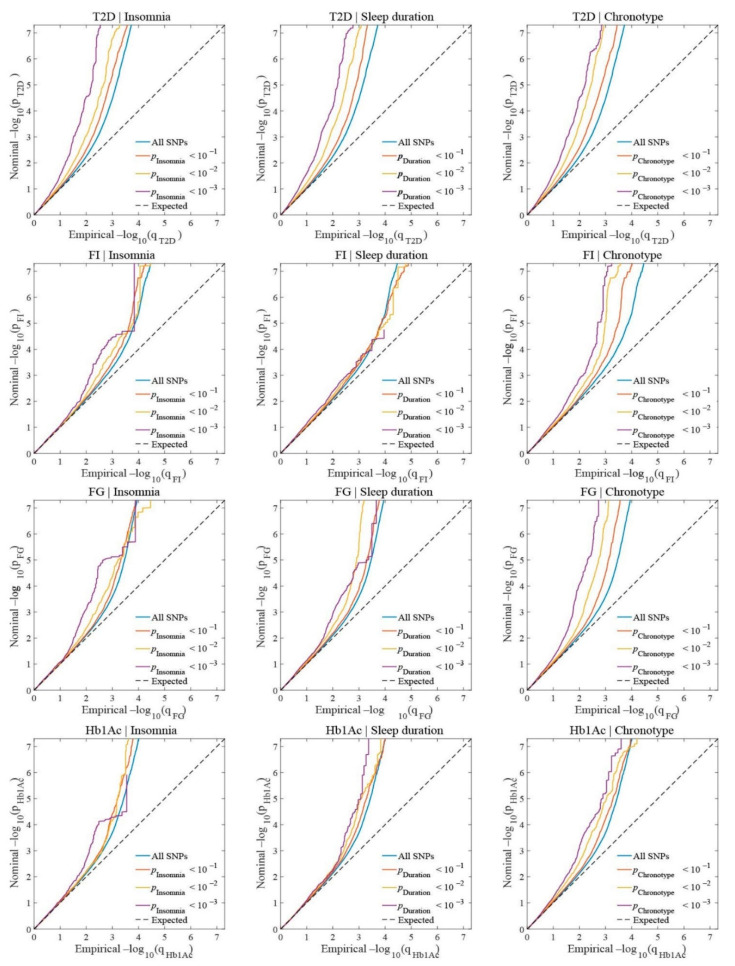
Polygenic overlap between diabetes-related traits (type 2 diabetes, fasting glucose, glycated hemoglobin, and fasting insulin) and sleep traits (insomnia symptoms, sleep duration, and chronotype). Conditional Q–Q plots of nominal versus empirical −log10 *p*-values (corrected for inflation) in type 2 diabetes, fasting glucose, glycated hemoglobin, and fasting insulin, below the standard genome-wide association study threshold of 5 × 10^−8^ as a function of significance of association with insomnia, sleep duration, and chronotype, respectively, at the level of −log10 (*p*) ≥ 1, −log10 (*p*) ≥ 2, and −log10 (*p*) ≥ 3, corresponding to *p* ≤ 0.10, *p* ≤ 0.01, and *p* ≤ 0.001, respectively. The blue lines indicate all SNPs. The dashed lines indicate the null hypothesis.

**Figure 2 biomedicines-10-00368-f002:**
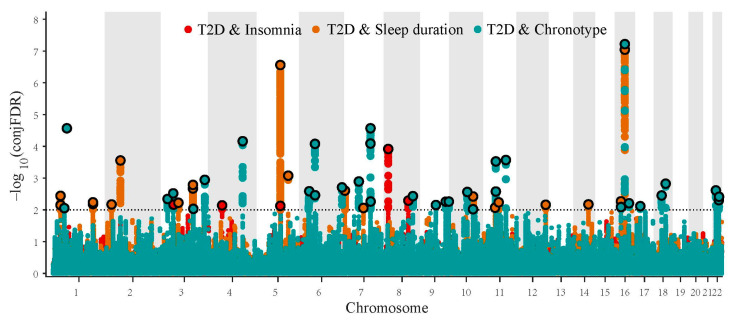
Conjunction FDR Manhattan plot of conjunctional –log10 (FDR) values for type 2 diabetes and insomnia, sleep duration, or chronotype. SNPs with conjunctional FDR < 0.01 (i.e., conjunctional –log10 (FDR) > 2.0) are shown with enlarged data points, whereas the small points represent other SNPs. A black circle around the enlarged data points indicates the most significant SNP in each LD block. The figures show the localization of the “conjunctional loci”, and further details are presented in Table 1.

**Table 1 biomedicines-10-00368-t001:** Genetic loci with conjunction FDR < 0.01 shared between type 2 diabetes and sleep traits.

Locus No.	SNP	Gene	Chr:Pos	A1/A2	Z Score Type 2 Diabetes	Z Score Sleep Traits	ConjFDR	*p*-Value Type 2 Diabetes	*p*-Value Sleep Traits
Type 2 diabetes and insomnia									
1	rs2820290	*NAV1*	1:201783682	A/G	−3.90	−4.62	0.01	9.69 × 10^−5^	3.88 × 10^−6^
1	rs2820290	*IPO9-AS1*	1:201783682	A/G	−3.90	−4.62	0.01	9.69 × 10^−5^	3.88 × 10^−6^
2	rs4688760	*RBM6*	3:49980596	C/T	−5.67	−4.42	0.01	1.44 × 10^−8^	9.82 × 10^−6^
3	rs67073213	Upstream: *SPATA18*; Downstream: *RP11-588F10.1*	4:53286872	A/G	3.87	−4.48	0.01	1.10 × 10^−4^	7.57 × 10^−6^
4	rs26434	*PAM*	5:102363402	C/T	5.90	−4.40	0.01	3.58 × 10^−9^	1.10× 10^−5^
5	rs4526367	*MSRA*	8:10213462	G/A	−4.96	5.49	0.00	7.10 × 10^−7^	3.99 × 10^−8^
6	rs4735334	*NDUFAF6*	8:95955292	G/A	3.98	4.63	0.01	6.91 × 10^−5^	3.70 × 10^−6^
6	rs4735334	*TP53INP1*	8:95955292	G/A	3.98	4.63	0.01	6.91 × 10^−5^	3.70 × 10^−6^
Type 2 diabetes and sleep duration									
7	rs4949329	*PUM1*	1:31440361	T/C	−4.04	4.78	0.00	5.29 × 10^−5^	1.80 × 10^−6^
8	rs61780511	Upstream: *PUM1*; Downstream: *SEPW1P*	1:31546006	G/A	3.83	−4.12	0.01	1.27 × 10^−4^	3.81 × 10^−5^
9	rs12137232	*LMOD1*	1:201885446	T/G	−3.90	4.36	0.01	9.69 × 10^−5^	1.30 × 10^−5^
10	rs6711622	*DNMT3A*	2:25531350	A/G	−3.85	4.14	0.01	1.19 × 10^−4^	3.51 × 10^−5^
11	rs1641155	*LINC01122*	2:58965211	G/T	4.69	4.90	0.00	2.75 × 10^−6^	9.36 × 10^−7^
12	rs12485697	Upstream: *RP11-231I13.2*; Downstream: *COX6CP6*	3:70543116	T/C	3.89	−4.16	0.01	1.01 × 10^−4^	3.15 × 10^−5^
13	rs9844666	*PCCB*	3:135974216	A/G	4.19	4.60	0.00	2.77 × 10^−5^	4.27 × 10^−6^
14	rs1291921	*PCCB*	3:136036226	A/G	−4.35	−4.50	0.00	1.36 × 10^−5^	6.85 × 10^−6^
15	rs11242483	*PAM*	5:102323766	T/C	6.07	6.28	0.00	1.31 × 10^−9^	3.39 × 10^−10^
16	rs329124	*JADE2*	5:133865452	G/A	5.14	−4.65	0.00	2.80 × 10^−7^	3.30 × 10^−6^
17	rs62442924	*MAD1L1*	7:1989976	T/C	−4.15	5.16	0.00	3.33 × 10^−5^	2.47 × 10^−7^
18	rs7790729	*AUTS2*	7:69598649	T/C	3.76	4.07	0.01	1.69 × 10^−4^	4.77 × 10^−5^
19	rs3121426	Upstream: *5-Mar*; Downstream: *MARK2P9*	10:94153435	T/G	−6.69	−4.29	0.00	2.19 × 10^−11^	1.82 × 10^−5^
20	rs11037564	*HSD17B12*	11:43708725	C/T	−3.77	5.01	0.01	1.65 × 10^−4^	5.41 × 10^−7^
21	rs174533	*MYRF*	11:61549025	A/G	−3.90	4.44	0.01	9.69 × 10^−5^	8.82 × 10^−6^
21	rs174533	*TMEM258*	11:61549025	A/G	−3.90	4.44	0.01	9.69 × 10^−5^	8.82 × 10^−6^
22	rs12820906	*PITPNM2*	12:123493123	G/A	−5.41	4.11	0.01	6.39 × 10^−8^	3.90 × 10^−5^
23	rs12433645	*NRXN3*	14:80028314	T/C	−4.30	−4.12	0.01	1.75 × 10^−5^	3.75 × 10^−5^
24	rs4780887	*PDILT*	16:20393562	C/A	−3.92	4.23	0.01	8.78 × 10^−5^	2.39 × 10^−5^
25	rs8047587	*FTO*	16:53798622	T/G	16.19	−6.66	0.00	6.29 × 10^−59^	2.66 × 10^−11^
Type 2 diabetes and chronotype									
26	rs148262742	Upstream: *CDKN2C*; Downstream: *MIR4421*	1:51472241	C/T	−5.34	−3.95	0.01	9.40 × 10^−8^	7.93 × 10^−5^
27	rs12140153	*INADL*	1:62579891	T/G	−5.18	−6.10	0.00	2.21 × 10^−7^	1.03 × 10^−9^
28	rs903518	*UBE2E2*	3:23336968	G/A	−3.93	−4.22	0.00	8.32 × 10^−5^	2.42 × 10^−5^
29	rs78580841	*CCDC12*	3:46986452	T/C	4.05	4.78	0.00	5.03 × 10^−5^	1.72 × 10^−6^
30	rs1679147	*MRAS*	3:138097537	A/G	4.45	−3.93	0.01	8.72 × 10^−6^	8.49 × 10^−5^
31	rs17774982	*ST6GAL1*	3:186684460	C/T	−4.88	−4.51	0.00	1.07 × 10^−6^	6.39 × 10^−6^
32	rs1296328	*RP11-775H9.2*	4:137083193	A/C	4.98	6.11	0.00	6.37 × 10^−7^	9.73 × 10^−10^
33	rs1265945	*EHMT2*	6:31861815	G/A	−4.10	4.71	0.00	4.15 × 10^−5^	2.51 × 10^−6^
34	rs734597	Upstream: *RPS17P5*; Downstream: *RP4-753D5.3*	6:50836279	A/G	6.06	5.12	0.00	1.35 × 10^−9^	2.99 × 10^−7^
35	rs4434471	Upstream: *FTH1P5*; Downstream: *RP3-437C15.2*	6:51146875	G/A	4.02	4.37	0.00	5.94 × 10^−5^	1.24 × 10^−5^
36	rs66930764	Upstream: *RP5-826L7.1*; Downstream: *RP1-230L10.1*	6:164103243	A/G	−5.02	−4.38	0.00	5.24 × 10^−7^	1.21 × 10^−5^
37	rs11555134	*GRB10*	7:50659193	T/C	4.30	−5.30	0.00	1.75 × 10^−5^	1.17 × 10^−7^
38	rs77655131	*ORAI2*	7:102086552	T/C	4.94	−5.54	0.00	7.70 × 10^−7^	3.03 × 10^−8^
39	rs11496066	*FBXL13*	7:102486254	C/T	−5.18	5.90	0.00	2.21 × 10^−7^	3.62 × 10^−9^
40	rs62482405	*PSMC2*	7:102987583	G/T	4.36	−4.09	0.01	1.29 × 10^−5^	4.23 × 10^−5^
41	rs3808478	*TRPS1*	8:116678277	C/T	−4.00	4.86	0.00	6.43 × 10^−5^	1.15 × 10^−6^
42	rs6559752	*C9orf64*	9:86570075	T/C	−3.80	4.17	0.01	1.44 × 10^−4^	3.09 × 10^−5^
43	rs6478623	*DENND1A*	9:126315123	G/T	3.88	−4.68	0.01	1.06 × 10^−4^	2.81 × 10^−6^
44	rs11145756	*SEC16A*	9:139364585	G/A	−4.63	4.09	0.01	3.64 × 10^−6^	4.24 × 10^−5^
45	rs10998304	*TET1*	10:70342775	C/T	4.08	−4.36	0.00	4.41 × 10^−5^	1.28 × 10^−5^
46	rs143539037	*CPEB3*	10:93827055	T/C	5.92	−3.92	0.01	3.26 × 10^−9^	8.80 × 10^−5^
47	rs11039307	Upstream: *FAM180B*; Downstream: *C1QTNF4*	11:47611152	T/C	5.22	4.83	0.00	1.77 × 10^−7^	1.37 × 10^−6^
48	rs11039358	*FNBP4*	11:47746962	G/A	4.52	4.30	0.00	6.10 × 10^−6^	1.72 × 10^−5^
49	rs4237555	Upstream: *MTNR1B*; Downstream: *RPL26P31*	11:92725803	C/T	−7.86	−4.85	0.00	3.84 × 10^−15^	1.22 × 10^−6^
50	rs4606726	*PDILT*	16:20383700	G/A	−3.83	−3.97	0.01	1.27 × 10^−4^	7.10 × 10^−5^
51	rs8047587	*FTO*	16:53798622	T/G	16.19	7.42	0.00	6.29 × 10^−59^	1.19 × 10^−13^
52	rs217184	*TXNL4B*	16:72105965	C/T	−3.98	4.06	0.01	6.91 × 10^−5^	4.99 × 10^−5^
52	rs217184	*HPR*	16:72105965	C/T	−3.98	4.06	0.01	6.91 × 10^−5^	4.99 × 10^−5^
53	rs3816511	*PEMT*	17:17409401	G/A	3.77	4.79	0.01	1.61 × 10^−4^	1.63 × 10^−6^
54	rs1371319	Upstream: *RP11-687D19.1*; Downstream: *RN7SKP182*	18:36277087	C/T	4.50	4.22	0.00	6.94 × 10^−6^	2.45 × 10^−5^
55	rs17596995	*TCF4*	18:53166594	A/G	−4.25	−4.94	0.00	2.12 × 10^−5^	7.66 × 10^−7^
56	rs5762622	*TTC28*	22:28835458	A/G	4.28	−4.32	0.00	1.87 × 10^−5^	1.57 × 10^−5^
57	rs5757906	*TNRC6B*	22:40687757	C/T	−3.91	−4.27	0.00	9.24 × 10^−5^	1.91 × 10^−5^
58	rs28741121	*XRCC6*	22:42025823	A/G	−4.07	−4.19	0.00	4.72 × 10^−5^	2.74 × 10^−5^

Abbreviations: SNP, single nucleotide polymorphisms; Chr, chromosome; Pos, position; A1, allele 1; A2, allele 2; FDR, false discovery rate.

**Table 2 biomedicines-10-00368-t002:** Causal relationship between gene expression and type 2 diabetes, insomnia, chronotype, and sleep duration.

Genes	Type 2 Diabetes				Insomnia				Chronotype			Sleep Duration		
	N_IV_	OR	95% CI	*p*	N_IV_	OR	95% CI	*p*	OR	95% CI	*p*	OR	95% CI	*p*
*ENSA*	2	1.10	1.10 (0.98–1.23)	0.1119	2	1.14	1.14 (1.05–1.24)	0.0013	0.96	0.96 (0.88–1.06)	0.4454	1.01	1.01 (0.97–1.05)	0.6732
*CPEB3*	1	1.43	1.43 (1.30–1.56)	0.0000										
*MYBPC3*	6	0.95	0.95 (0.92–0.99)	0.0173	5	1.02	1.02 (0.98–1.06)	0.2920	1.00	1.00 (0.98–1.01)	0.5591	0.99	0.99 (0.98–1.00)	0.0880
*MYRF*	1	0.94	0.94 (0.90–0.98)	0.0049	1	1.03	1.03 (1.00–1.07)	0.0581	1.00	1.00 (0.98–1.01)	0.8016	1.01	1.01 (1.00–1.03)	0.1302
*KLHL29*	2	0.99	0.99 (0.91–1.07)	0.7417	2	0.89	0.89 (0.83–0.95)	0.0008	1.00	1.00 (0.97–1.03)	0.9629	1.03	1.03 (0.99–1.06)	0.1057
*DNMT3A*	1	1.07	1.07 (0.98–1.15)	0.1161										
*XRCC6*	2	1.06	1.06 (0.91–1.23)	0.4503	2	1.03	1.03 (1.00–1.07)	0.0469	0.97	0.97 (0.96–0.99)	0.0002	1.01	1.01 (0.99–1.03)	0.1748
*PCCB*	3	1.03	1.03 (0.96–1.1)	0.4567	3	1.00	1.00 (0.97–1.02)	0.8452	1.01	1.01 (0.99–1.02)	0.3804	1.00	1.00 (0.96–1.03)	0.8049
*MAD1L1*	4	1.00	1.00 (0.97–1.04)	0.8834	4	1.02	1.02 (0.99–1.04)	0.1612	1.00	1(0.98–1.02)	0.7269	0.99	0.99 (0.97–1.00)	0.0167
*PMPCA*	1	0.74	0.74 (0.62–0.87)	0.0003	1	0.93	0.93 (0.82–1.07)	0.3157	1.15	1.15 (1.08–1.22)	0.0000	0.99	0.99 (0.93–1.06)	0.7955
*INPP5E*	4	1.10	1.10 (1.07–1.13)	0.0000	4	1.00	1.00 (0.98–1.02)	0.9185	0.98	0.98 (0.97–0.99)	0.0001	1.00	1.00 (0.99–1.02)	0.8376
*SEC16A*	2	1.08	1.08 (1.05–1.12)	0.0000										

N_IV_, number of instrumental variables; OR, odds ratio; CI, confidence interval; *p*, strength of evidence against the null hypothesis of no association between variant and outcome.

## Data Availability

Our datasets analyzed during the current study were derived from the following public domain resources: Summary statistics of type 2 diabetes is available from DIAGRAM consortium (http://diagram-consortium.org/, accessed on 13 November 2020) and summary statistics of the diabetes-related traits are available from Meta-Analyses of Glucose and Insulin-related traits Consortium (https://magicinvestigators.org/, accessed on 13 November 2020). Summary-level data for sleep traits are available at https://ctg.cncr.nl/software/summary_statistics (accessed on 13 November 2020).

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
