# Peer review of "Integrative Identification of Genetic Loci Jointly Influencing Diabetes-Related Traits and Sleep Traits of Insomnia, Sleep Duration, and Chronotypes"

_biomedicines, 2022, doi:10.3390/biomedicines10020368_

Round 1

Reviewer 1 Report

In this study, the authors identified polygenic risk variants shared between traits of type 2 diabetes (T2D) and sleep. For T2D, these traits included risk of T2D, elevated fasting glucose, fasting insulin and glycated hemoglobin, and for sleep, these included insomnia symptoms, sleep duration, and chronotype. From genome-wide association studies (GWAS), the authors used conditional quantile-quantile plots to evaluate pleiotropic enrichment between T2D and sleep traits. Next, the authors sought to elucidate the biology behind these shared polygenic risk variants using functional annotation and expression analysis. Finally, causality was ascertained using Mendelian randomization analysis.

The paper was well written and the various analyses were meticulously performed. Results were presented in a clear and coherent manner. I am therefore delighted to offer only minor comments stemming from scientific curiosity.

Comment 1: Will the shared polygenic risk variants differ by sex, given that sex has been associated with risk of T2D and various aspects of sleep?

Comment 2: Do the authors expect differences in results for persons of other ethnicities? The UK Biobank, from which the GWAS for sleep traits were obtained, consists largely of participants of European ancestry (94.6% [1]). However, in Mahajan et al.’s work where the T2D traits were obtained, only 70% of the participants were of European descent. Will this be a concern?

Comment 3: The condFDR approach is known to be conservative. Have the authors considered less conservative methods to assess cross-trait pleiotropy, such as genetic correlation, and checked if results differed substantially?

Comment 4: The sleep traits (insomnia symptoms, sleep duration, and chronotype) were each assessed at baseline using only one self-reported measure. In contrast, and using insomnia as an example, insomnia may be assessed using the Insomnia Severity Index that consists of seven questions [2]. Are the authors concerned about measurement errors? Have the questions on sleep been validated? This is a limitation that the authors may discuss in greater detail.

Comment 5: The authors’ finding that the genes PMPCA and INPP5E influencing chronotype were associated with risk of T2D is particularly interesting, given conflicting results in literature. I agree that the discrepancies may have been due to confounding by other factors. Since T2D status is available in the UK Biobank, have the authors conducted their own association analysis between chronotype and T2D risk, adjusting for potential confounders such as age and BMI?

Comment 6: In line 315, should ‘odds’ be ‘odd’ instead?

  1. Fry, A., et al., Comparison of Sociodemographic and Health-Related Characteristics of UK Biobank Participants With Those of the General Population. American Journal of Epidemiology, 2017. 186(9): p. 1026-1034.
  2. Morin, C.M., et al., The Insomnia Severity Index: psychometric indicators to detect insomnia cases and evaluate treatment response. Sleep, 2011. 34(5): p. 601-608.

Author Response

Thanks a lot for the valuable comments. The detailed point-by-point responses have been uploaded as a Word file. Please see the attachment.

Reviewer 2 Report

The manuscript presented for review examined the genetic basis of type 2 diabetes and sleep traits. The data used in the analysis came from genome-wide association studies.
The manuscript is clearly written, interesting, and presents valuable results.

However, I have slight comments:

- the introduction lacks a definition of insomnia and a chronotype
- poor diagnosis of sleep disorders and chronotype should be better discussed as a limitation of this analysis. One question does not diagnose insomnia. And a time criterion is necessary for its diagnosis. In addition, most patients suffer from secondary insomnia than primary. Because of these limitations, "insomnia symptoms" rather than "insomnia" should be used in the text.
- there are no simple conclusions from the study: that e.g. "people prone to be more an 'evening' than a 'morning' person have a higher risk for developing type 2 diabetes"
- due to a very poor sleep assessment, the sentence in the conclusion "Our study provided comprehensive evidence" is inappropriate

- line 276, should be "traits"

Author Response

(The authors gave the same response as above.)
